# Bovine Serum Albumin Interaction with Polyanionic and Polycationic Brushes: The Case Theoretical Study

**DOI:** 10.3390/ijms24043395

**Published:** 2023-02-08

**Authors:** Tatiana O. Salamatova, Ekaterina B. Zhulina, Oleg V. Borisov

**Affiliations:** 1Chemical Engineering Center, ITMO University, 197101 St. Petersburg, Russia; 2Institute of Macromolecular Compounds of the Russian Academy of Sciences, 199004 St. Petersburg, Russia; 3CNRS, Université de Pau et des Pays de l’Adour UMR 5254, Institut des Sciences Analytiques et de Physico-Chimie pour l’Environnement et les Matériaux, 64053 Pau, France

**Keywords:** polyelectrolyte brushes, protein absorption, polyampholytes, bionanocolloids

## Abstract

We apply a coarse-grained self-consistent field Poisson-Boltzmann framework to study interaction between Bovine Serum Albumin (BSA) and a planar polyelectropyte brush. Both cases of negatively (polyanionic) and positively (polycationic) charged brushes are considered. Our theoretical model accounts for (1) re-ionization free energy of the amino acid residues upon protein insertion into the brush; (2) osmotic force repelling the protein globule from the brush; (3) hydrophobic interactions between non-polar areas on the globule surface and the brush-forming chains. We demonstrate that calculated position-dependent insertion free energy exhibits different patterns, corresponding to either thermodynamically favourable BSA absorption in the brush or thermodynamically or kinetically hindered absorption (expulsion) depending on the pH and ionic strength of the solution. The theory predicts that due to the re-ionization of BSA within the brush, a polyanionic brush can efficiently absorb BSA over a wider pH range on the “wrong side” of the isoelectric point (IEP) compared to a polycationic brush. The results of our theoretical analysis correlate with available experimental data and thus validate the developed model for prediction of the interaction patterns for various globular proteins with polyelectrolyte brushes.

## 1. Introduction

Interactions of globular proteins with polyelectrolyte brushes (layers of charged macromolecules end-attached to a planar substrate or to the surface of colloidal particles and immersed in an aqueous solution) have been extensively studied both experimentally [1,2,3,4] and theoretically [5,6,7,8,9,10] in the past two decades.

The motivation to study the interactions between globular proteins and charged colloidal polymer nanostructures is twofold: First, such a system mimics extracellular structures of strongly charged natural polyelectrolytes, e.g., glycosaminoglycans (GAGs) on the surface of cells; their interactions with proteins are highly important for many biological processes [11,12,13,14,15]. Profound understanding of the role of non-specific electrostatic interaction between natural polyelectrolytes and proteins has thus fundamental scientific importance. Second, colloidal polyelectrolyte brushes, microgels, or micelles with polyelectrolyte coronae are actively explored for biomedical applications such as drug and gene delivery, inhibition of viral infection, etc. [16,17,18,19,20,21,22,23,24,25].

Globular proteins can be assimilated to nanocolloidal particles bearing weak (pH-sensitive) acidic and basic groups (amino acid residues) on their surface exposed to the surrounding aqueous environment. Hence, globular proteins can be termed as weak polyampholytes. In contrast to “strong” polyampholytes, which comprise quenched arrays of positively and negatively charged groups, the distribution of positive and negative charges on the surface of the weak polyampholytic nanocolloidal particles and, throughout, its dipole moment and net charge adjust to the local electrostatic potential that controls local pH. Moreover, even the sign of the net charged by the weak polyampholyte can change depending on local electrostatic potential (the so-called charge inversion phenomenon).

The PE brush, as an array of heavily charged macroions attached to the surface, creates a strong electrostatic field, which promotes re-ionization of the weak polyampholyte, up to the charge inversion. This charge inversion has been proposed as one of the driving forces for absorption of globular proteins on the “wrong side” of the isoelectric point (IEP), that is, above IEP for polycationic and below IEP for the polycationic brush [5,6]. The charge–charge correlation effects, including electrostatically-driven adsorption of the segments of the brush-forming PE chains onto oppositely charged “patches” on the globule surface, were involved as a possible explanation for the protein absorption by the brush on the “wrong side” of the IEP as well [8,9,10]. Both mechanisms are consistent with the experimental observation that such absorption is efficiently suppressed by increasing the salt concentration in the solution [1,2].

A comprehensive set of experimental data was accumulated for one of the most common globular proteins, such as Bovine Serum Albumin (BSA) interacting with either polyanionic or polycationic brushes: Selective absorption of BSA and β -glucosidase (β-G) by cationic “annealed” (pH-sensitive) and “quenched” (strong) spherical polyelectrolyte brushes (SPB) was systematically studied in ref. [26]. Two types of brushes consisting of the same polystyrene core and a shell of poly (2-aminoethylmethacrylate chloride) (PAEMH) and poly [2-(methacryloyloxy) ethyl] trimethylammonium chloride (PMAETA), respectively, were used. It was found that the absorption of the protein by the brush is controlled by the concentration of salt and pH of the buffer and at low ionic strength, absorption in quenched brushes occurred better than in annealed ones. The protein-excluded volume effect was also investigated and it was determined using the small-angle X-ray scattering (SAXS) analysis, that larger proteins are most likely absorbed on the outer layer of the brush. The absorption of Bovine Hemoglobin (BHb) near the isoelectric point (at pH = 7.2) on quenched spherical brushes of poly (styrene sulfonic acid) obtained by photoemulsion polymerization from the polystyrene core was studied in ref. [27]. The location of hemoglobin molecules inside the brush was determined by using SAXS, and changes in the secondary structure of the protein inside the SPBs were determined using Fourier transform infrared spectroscopy (FTIR). It was found that the proteins penetrate deep into the brush and about 30% of the proteins are absorbed on the surface of the core due to hydrophobic interactions, while the rest is closely associated with polyelectrolyte chains. Thus, it was concluded that there is no steric penalty for BHb penetration into the brush, despite the tightly packed polyelectrolyte layer. These observations differ from those obtained by Ballauff and colleagues in refs. [2,28], where the interaction of BSA and Bovine Pancreatic Ribonuclease A (RNase A) with SPB, which consists of a solid poly(styrene) core with a diameter of about 100 nm with long densely grafted [poly(styrene sulfonic acid, PSS) or poly (acrylic acid, PAA)] polyelectrolyte chains, was studied. It was found that the proteins are also strongly absorbed at low ionic strength of the solution, as in the work [27], even when they carry the charge of the same sign. However, unlike BHb, proteins are evenly distributed along polyelectrolyte chains and avoid direct contact with the hydrophobic surface (except in cases of strong absorption). Note that at high ionic strength, absorption practically does not occur on the “wrong side” of the isoelectric point.

Advances in the theory of PE brushes enabled us to obtain analytical expressions for the distribution of polymer density and electrostatic potential (and thus concentrations of mobile ions) inside the PE brush and in the solution in the proximity of the brush on the basis of self-consistent field Poisson-Boltzmann approximation [29,30,31].

The aim of the present paper is to use the Poisson-Boltzmann framework to evaluate the position-dependent free energy and net charge of the BSA modeled as a polyampholytic nanocpaticle inserted into polyanionic or polycationic brush. In our analysis, we account for the contributions of ionic as well as short-range attractive interactions between the surface of the BSA globule and the brush-forming chains. Hence, our major goal is to unravel cooperative or competitive effects of non-specific fundamental (electrostatic, hydrophobic) interactions on the BSA uptake by either polyanionic or polycationic brushes.

In particular, analysis of the free energy profiles enable to identify under which conditions electrostatically or hydrophobically-driven spontaneous uptake of BSA by the brush may occur. Particular attention is paid to the protein-brush interaction on the “wrong side” of the IEP (that is, at pH above or below IEP in the cases of polyanionic and polycationic brushes, respectively), when the BSA in the buffer and the PE brush carry the net charge of the same sign.

The rest of the paper is organized as follows: In Section 2 “Results and Discussion” we first describe our theoretical model (Section 2.1). In Section 2.2 we introduce dominant (re-ionization, osmotic and short-range non-electrostatic) contributions to the free energy of the protein insertion into the PE brush. The BSA insertion free energy profiles calculated under varied conditions for anionic and cationic brushes are presented in Section 2.3. The conclusions are formulated in Section 4. A brief summary of the Poisson-Boltzmann theory of PE brushes, electrostatic potential and monomer density distributions inside the brush and in the exterior solution in the proximity of the brush is given in the “Materials and Methods” Section 3.

## 2. Results and Discussion

### 2.1. Model

Within our coarse-grained approach we model the BSA globule as a nanocolloidal particle with volume V=169.843 nm${}^{3}$ and total surface area A=567.92 nm${}^{2}$ [32].

The crystal structure of the BSA published by Majorek et al. [32] is shown in Figure 1. The BSA molecule consists of 583 amino acid residues connected into one chain (A), which is linked by 17 cystine residues. The chain is formed by three homologous, but structurally different domains: I, II, III, which, in turn, are subdivided into sub-domains A and B. In general, a protein molecule has the shape of a heart. Ionized residues give the protein a high total charge [33].

There are several types of cationic and anionic ionogenic groups localized at the globule-water interface, each characterized by specific ionization (via protonation or deprotonation) constant $K_{i\pm }$. We ascribe respective ionization constants $K_{i-}$ to $N_{i-}$ anionic groups of type (i−=1,2,…) and the ionization constants $K_{i+}$ to $N_{i+}$ cationic groups of type (i+=1,2,…). The numbers of the amino acid residues of each type and the respective ionization constants are collected in Table 1. The area of the globule-water surface occupied by apolar amino acid residues is estimated as $A^{\prime }=322.72$ nm${}^{2}$ [34].

The globule is interacting with the polyelectrolyte brush, Figure 2, immersed in the aqueous solution. The latter is treated as a continuous medium with dielectric permittivity ϵ. The brush is formed by polyelectrolyte (polyacid or polybase) chains with the degree of polymerization N≫1 end-tethered with density σ, where $\sigma =a^{2}/s$ and *s* is the grafting area per chain, to the planar surface localized at z=0. The polyelectrolyte chains are assumed to be intrinsically flexible (the monomer unit length is on the order of Kuhn segment *a*). Below we express all the lengths and distances in *a* units. We assume the fraction α of negatively or positively charged monomer units to be quenched (pH-independent). The solution contains monovalent (positively and negatively charged) mobile ions of low molecular weight salt with concentration (number density).
(1)$$c_{b+}=c_{b-}=c_{s}$$

The concentration of hydrogen ions $\left[H^{+}\right]$ and thus pH in the bulk of the solution are fixed to the values of $pH_{b}=-log_{10}\left[H_{b}^{+}\right]$. The polyelectrolyte brush gives rise to *z*-dependent electrostatic potential Ψ(z), which vanishes at z→∞, that is, far away from the brush in the bulk of the solution. The local concentration of hydrogen ions at distance *z* from the grafting surface can be expressed as
(2)$$\left[H^{+}\left(z\right)\right]={\left[H^{+}\right]}_{b}\exp\left(-\psi \left(z\right)\right)$$
where $\psi \left(z\right)\equiv e\Psi \left(z\right)/k_{B}T$ is reduced (dimensionless) electrostatic potential, which is presented in the explicit form in the Materials and Methods section. Here *e* is the elementary charge, $k_{B}$ is the Boltzmann constant and *T* is the temperature. Hence, the concentration of hydrogen ions in the polyanionic/polycationic brush is larger/smaller than in the bulk of the solution.

Then the pH-(and position)-dependent net charge of the globule is given by
(3)$$Q\left(z\right)=\sum_{i+}^{}N_{i+}\alpha_{i+}\left(z\right)-\sum_{i-}^{}N_{i-}\alpha_{i-}\left(z\right)$$
where the ionization degrees of cationic and anionic groups depend on local pH as
$$\alpha_{i+}\left(z\right)={\left(1+K_{i+}/\left[H^{+}\left(z\right)\right]\right)}^{-1}\equiv$$
(4)$${\left(1+10^{pH\left(z\right)-pK_{i+}}\right)}^{-1}$$
and
$$\alpha_{i-}\left(z\right)={\left(1+\left[H^{+}\left(z\right)\right]/K_{i-}\right)}^{-1}\equiv$$
(5)$${\left(1+10^{pK_{i-}-pH\left(z\right)}\right)}^{-1},$$
respectively, and we use notation
(6)$$pH\left(z\right)\equiv -log_{10}\left[H^{+}\left(z\right)\right].$$

In the isoelectric point, $pH_{b}=pI$, the globule charge in the bulk of the solution $Q_{b}$ vanishes, that is,
(7)$$Q_{b}{|}_{pH_{b}=pI}={\left(\sum_{i+}^{}N_{i+}\alpha_{ib+}-\sum_{i-}^{}N_{i-}\alpha_{ib-}\right)}_{pH_{b}=pI}=0$$
where the respective ionization degrees of cationic and anionic residues in the bulk of the solution are given by
(8)$$\alpha_{bi+}={\left(1+K_{i+}/{\left[H^{+}\right]}_{b}\right)}^{-1}={\left(1+10^{pH_{b}-pK_{i+}}\right)}^{-1}$$
(9)$$\alpha_{bi-}={\left(1+{\left[H^{+}\right]}_{b}/K_{i-}\right)}^{-1}={\left(1+10^{pK_{i-}-pH_{b}}\right)}^{-1}$$

### 2.2. Insertion Free Energy

The position-dependent free energy of the protein globule ΔF(z), with the reference state in the bulk of the solution ΔF(z=∞)=0, is presented as
(10)$$\Delta F\left(z\right)=\Delta F_{ionic}\left(z\right)+\Delta F_{vol}\left(z\right)+\Delta F_{surf}\left(z\right)$$

The insertion of the globule from the outer solution into the brush is accompanied by re-ionization (change in the ionization states of cationic and anionic residues) that gives rise to the first term, $\Delta F_{ion}\left(z\right)$ in Equation (Equation 10). Following ref. [7], we present it as
(11)$$\Delta F_{ion}\left(z\right)/k_{B}T=\sum_{i+}^{}N_{i+}\ln\left(\frac{1-\alpha_{i+}\left(z\right)}{1-\alpha_{bi+}}\right)+\sum_{i-}^{}N_{i-}\ln\left(\frac{1-\alpha_{i-}\left(z\right)}{1-\alpha_{bi-}}\right)$$
where $\alpha_{i+}\left(z\right)$, $\alpha_{bi+}$, $\alpha_{i-}\left(z\right)$, $\alpha_{bi-}$ are the respective degrees of ionization of basic and acidic monomer units of type i± in the globule placed at distance *z* from the grafting surface or in the bulk of the solution (at z=∞). Because concentration of hydrogen ions in the polyanionic/polycationic brush is larger/smaller than in the bulk of the solution, acidic/basic residues become more strongly/weakly ionized upon insertion of the globule into polyacidic brush and vice versa for the polybasic brush. Therefore, the term $\Delta F_{ionic}\left(z\right)$ exhibits different dependence on $\Delta pH=pH_{b}-pI$ in the cases of polyanionic and polycationic brushes.

The electrostatic potential of the brush produces also an excess concentration of mobile counterions inside the brush and in the vicinity of the brush, thus giving rise to the excess osmotic pressure. Insertion of the globule into the brush is accompanied by the work performed against this osmotic pressure, which is described by the second term in Equation (Equation 10) and given by
(12)$$\Delta F_{vol}\left(z\right)=V\Delta \Pi \left(z\right)=V\left(c_{+}\left(z\right)+c_{-}\left(z\right)-2c_{s}\right),$$
which, with the account of the Boltzmann law for distribution of mobile ions
(13)$$c_{\pm }\left(z\right)=c_{s}\exp\left(\mp \psi \left(z\right)\right),$$
leads to
(14)$$\Delta F_{vol}\left(z\right)/k_{B}T=4Vc_{s}\sinh^{2}\left(\psi \left(z\right)/2\right).$$

Notably $\Delta F_{ion}\left(z\right)$ can vary non-monotonically and change its sign as a function of *z*, whereas $\Delta F_{vol}\left(z\right)$ is non-negative and a monotonously decreasing function of *z*, providing a thermodynamic force expelling the globule from the brush; both terms do not vanish at the brush edge, i.e., at z=0, but rather are operative in the double electrical layer protruding beyond the edge of the brush in the solution. Because the profiles of ΔΠ(z) are identical for polyanionic and polycationic brush, the term $\Delta F_{vol}\left(z\right)$ is also the same in these two cases.

Finally, we introduce the term $\Delta F_{surf}\left(z\right)$ describing non-electrostatic (e.g., hydrophobic) interactions between monomer units of the brush-forming chains and apolar areas (residues) exposed to the globule-solvent interface. Following a semi-empirical approach developed in ref. [36], we approximate this term as
(15)$$\Delta F_{surf}\left(z\right)/k_{B}T=\gamma \left\{c_{p}\left(z\right)\right\}\cdot A^{\prime }$$
where $A^{\prime }\leq A$ is the area of the apolar globule surface and
(16)$$\gamma \left\{c_{p}\left(z\right)\right\}=\Delta \chi_{ads}c_{p}\left(z\right)$$
where $c_{p}\left(z\right)$ is the concentration (volume fraction) of monomer units of the brush-forming PE chains at distance *z* from the grafting surface. We used an open source C library for solvent accessible surface area (SASA) calculations—FreeSASA to calculate the available area of apolar groups $A^{\prime }$ on the surface of the PDB structure of the protein “4F5S” [34]. We used approximation to calculate SASA by Lee and Richards (L&R) where the surface is approximated by the outline of a set of slices. The prefactor $\Delta \chi_{ads}$ quantifies (in $k_{B}T$ units) the differential contact free energy of the short-range interactions between monomer unit of the chain and the apolar area on the globule surface, with the account of the conformational entropy losses imposed by the presence of the impermeable for the chain surface of the globule. A negative value of $\Delta \chi_{ads}=-0.45$, used below, corresponds to moderately strong adsorption of the polymer on apolar areas of the globule interface with an adsorbed layer thickness on the order of a few monomer units. We remark that in the depletion case of $\Delta \chi_{ads}\geq 0$, the effect of short-range interactions on the protein-brush interactions is negligible, except under particular conditions of mutual compensation of $\Delta F_{ion}$ and $\Delta F_{vol}$, which is not considered here. Obviously, the term $\Delta F_{surf}\left(z\right)$ in the insertion free energy is independent of the sign of the charge of the brush-forming chains, i.e., is the same for polyanionic and for polycationic brushes.

### 2.3. Ionic Contribution to the Free Energy

We start with analyzing the patterns of the position-dependent re-ionization free energy, $\Delta F_{ion}\left(z\right)$. As demonstrated below, depending on the environmental conditions ($pH_{b}$ and salt concentration), this term in the free energy provides either electrostatic attractive force driving BSA absorption in the polyelectrolyte brush or repulsive force leading to the BSA exclusion from the brush.

In Figure 3a,b the insertion free energy $\Delta F_{ion}\left(z\right)$ and the net charge of the globule Q(z) are plotted at $pH_{b}=pI$, by solid and dashed lines for polyanionic and polycationic brushes, respectively, as a function of distance *z* from the grafting surface. In both cases the free energy monotonously decreases upon approaching the grafting surface (upon a decrease in *z*) and exhibits an edge minimum with a negative value at z=0, whereas the globule net charge grows in the absolute value from zero in the buffer, z=∞, and reaches a positive (in a polyanionic brush) or a negative (in a polycationic brush) value at the grafting surface. Both $|\Delta F_{ion}\left(z=0\right)|$ and |Q(z)| decrease if salt concentration increases. Hence, being enhanced by interaction with the brush ionization of oppositely (with respect to the brush) charged groups and suppressed ionization of the similarly charged groups on BSA globule surface gives rise to the electrostatic driving force of the BSA absorption by the brush in the IEP. The free energy $\Delta F_{ion}\left(z\right)$ and the net charge Q(z) exhibit similar patterns when the BSA in the buffer is charged oppositely to the brush, that is, at $pH<pI,Q_{b}>0$ or $pH>pI,Q_{b}<0$ in the cases of polyanionic and polycationic brushes, respectively. Notably, at pH=pI the depth of the minimum in $\Delta F_{ion}\left(z\right)$ and the absolute value of the BSA charge |Q(z=0)| are significantly larger in the case of the polyanionic brush compared to the polycationic one. This important observation implies that BSA near IEP is more strongly absorbed by the polyanionic brush rather than by the polycationic one.

The same trends can be observed in Figure 4a,b where we present 2D plots of ionic part of the position-dependent insertion free energy $\Delta F_{ion}(z,|\Delta pH_{b}\left|\right)$ and net charge $Q(z,|\Delta pH_{b}|)$, respectively, at varied salt concentrations for the cases of the polyanionic and polycationic brush. Here, $\Delta pH_{b}=pH_{b}-pI$, and $pH_{b}>pI$ in the case of the polyanionic brush (BSA is charged negatively in the buffer), whereas $pH_{b}<pI$ in the case of polycationic brush (BSA is charged positively in the buffer), that is, in both cases pH in the buffer corresponds to the “wrong side” of the IEP for the protein.

A new feature observed at $|\Delta pH_{b}|\geq 0$ compared to the IEP ( $\Delta pH_{b}=0$) is the non-monotonous character and appearance of a maximum in $\Delta F_{ion}\left(z\right)$ curves. This maximum (a potential barrier) is located at $z=z^{*}$ close to the edge of the brush and separates the exterior region, $z\geq z^{*}$, where $\partial \Delta F_{ion}\left(z\right)/\partial z<0$ from the proximal region (a potential well), where $\partial \Delta F_{ion}\left(z\right)/\partial z>0$. As one can see from Figure 4b, and can be demonstrated analytically, the position $z=z^{*}$ of the maximum in $\Delta F_{ion}\left(z\right)$ coincides with the point of the globule charge inversion, that is
$${\left(\frac{\partial \Delta F_{ion}\left(z\right)}{\partial z}\right)}_{z=z^{*}}=0,Q\left(z=z^{*}\right)=0.$$

Hence, the BSA globule negatively charged in the buffer, $Q_{b}<0,pH_{b}>pI$, acquires a positive charge inside the polyanionic brush at $z<z^{*}$, whereas the BSA globule positively charged in the buffer, $Q_{b}>0,pH_{b}<pI$, acquires a negative charge inside the polycationic brush at $z<z^{*}$. Obviously, in the charge inversion point, $z=z^{*}$, local $pH(z=z^{*})$ defined by Equation (Equation 6) is equal to pI, both in the polyanionic as well as in the polycationic brush. Therefore, for the same $|\Delta pH_{b}|$, the positions $z=z^{*}$ of the charge inversion points (and maxima in $\Delta F_{ion}\left(z\right)$) coincide in the polyanionic and in the polycationic brushes, as one can easily see in Figure 4b.

An increase in $|\Delta pH_{b}|$ or in salt concentration results in shrinking and decrease in the depth $|\Delta F_{ion}\left(z=0\right)|$ of the proximal potential well. The potential barrier is shifted towards the grafting surface (a decrease in $z^{*}$) with a concomitant increase in its height, $|\Delta F_{ion}\left(z=z^{*}\right)|$. Therefore, even if the charge of BSA changes the sign at $z=z^{*}$ (the BSA charge inversion in the brush occurs) and $\Delta F_{ion}\left(z=0\right)<0$, the absorption can be hindered kinetically: the BSA globule cannot overcome the electrostatic potential barrier and enter the brush.

Moreover, the globule charge inversion at $z=z^{*}$ does not necessarily imply that $\Delta F_{ion}\left(z=0\right)<0$. As one can see in Figure 4, the ionic part of the free energy may exhibit a local edge minimum at z=0 with $\Delta F_{ion}\left(z=0\right)>0$, which is separated by the potential barrier at $z=z^{*}$ from the exterior solution where $\Delta F_{ion}\left(z=\infty \right)=0$. In this case the position of the globule charged oppositely with respect to the brush inside the brush corresponds to the metastable and not to the equilibrium state.

As follows from analysis of Figure 4a, both an increase in $C_{s}$ at $|\Delta pH_{b}|\geq 0$ and an increase in $|\Delta pH_{b}|$ at constant $C_{s}$ hinders and eventually suppressed absorption of the BSA by the similarly charged PE brush: The $\Delta F_{ion}\left(z\right)$ becomes a monotonously decreasing function of *z* that corresponds to the globule expulsion from the brush. This result is perfectly in line with experimental observations. Remarkably, the range of ($|\Delta pH_{b}|,C_{s}$), where such absorption is thermodynamically driven, is noticeably wider for the polyanionic than for the polycationic brush.

### 2.4. Osmotic and Non-Electrostatic Contributions to the Free Energy

According to Equations (Equation 14) and (Equation 15), the osmotic, $\Delta F_{vol}\left(z\right)$ and non-electrostatic, $\Delta F_{surf}\left(z\right)$, contributions to the free energy are independent of the ionization state of the BSA, but depend on the brush properties, that is, the excess osmotic pressure in the brush, which is related to the absolute value of the electrostatic potential ψ(z), and distribution of the monomer density $c_{p}\left(z\right)$ in the *z*-direction. Therefore, these terms in the free energy do not depend on $pH_{b}$, but depend on salt concentration, which affects both ψ(z) and $c_{p}\left(z\right)$. In Figure 5 and Figure 6, we present 2D profiles of $\Delta F_{vol}\left(z,C_{s}\right)$ and $\Delta F_{surf}\left(z,C_{s}\right)$ 2D, which are identical for polyanionic and polycationic brushes.

As one can see from Figure 5, an increase in salt concentration leads to simultaneous contraction of the brush (decrease in *H*) and the decrease in the magnitude of the osmotic repulsive term $\Delta F_{vol}\left(z\right)$ due to the decrease in the excess osmotic pressure in the brush.

On the contrary, contraction of the brush upon an increase in $C_{s}$ leads to a magnification of the attractive $\Delta F_{surf}\left(z\right)$ term, which grows proportionally to the polymer density in the brush $c_{p}\left(z\right)$, as seen in Figure 6. Hence, an increase in salt concentration provokes the deepening of the edge minimum in $\Delta F_{surf}\left(z\right)$ with a concomitant decrease in its width.

We remark, that in a real experimental situation, variations in both $pH_{b}$ and $C_{s}$ may also affect the conformation of the BSA globule and thus lead to variations in the globule volume *V* and surface area *A*. We do not, however, account for these relatively weak effects in our model; although this can be readily implemented by using empirical dependences $V(pH,c_{s})$ and $A(pH,c_{s})$ in Equations (Equation 14) and (Equation 15).

### 2.5. The Net Insertion Free Energy

The 2D profiles of the net free energy $\Delta F(z,|\Delta pH_{b}|)$ of the BSA insertion into polyanionic and polycationic brushes comprising the three contributions discussed above and defined by Equation (Equation 10) are presented in Figure 7a and Figure 7b, respectively. The crossections of the 2D profiles are shown for the same deviations from the IEP (above or below the IEP in the cases of the polyaionic and the polycationic brush, respectively).

As has been already mentioned, as a general trend, at the same deviation $|\Delta pH_{b}|$ from the IEP and the same salt concentration $C_{s}$, the polyanionic brush more strongly absorbs BSA than the polycationic one. However, comparison of Figure 7a,b indicates that even the shapes of the $\Delta F(z,|\Delta pH_{b}|)$ curves and their evolution upon variation of salt concentration are essentially different in the cases of polyanionic and polycationic brushes. At low salt concentration, the $\Delta F(z,|\Delta pH_{b}|)$ for the polyanionic brush is strongly dominated by ionic contribution $\Delta F_{ion}\left(z\right)$ and exhibits wide edge minimum at z=0 and a relatively weak maximum, the position of which approximately coincides with the charge reversal point. Hence, BSA spontaneous absorption by the polyanionic brush above the IEP is strongly electrostatically driven and not kinetically hindered. An increase in salt concentration suppresses the electrostatic driving force, but enhances short-range attraction due to the brush contraction and an increase in the average polymer concentration in the brush. As a result, a potential well appears in the central region of the brush, its depth increases and the position is moved towards the grafting surface as the salt concentration keeps increasing.

The shapes of the insertion free energy $\Delta F(z,|\Delta pH_{b}|)$ profiles are different in the case of the polycationic brush. Only in the immediate vicinity of the IEP, a weak maximum at the edge and a shallow minimum in the central region of the brush emerge at low salt concentration because of the BSA re-ionization, while close to the grafting surface $\Delta F(z,|\Delta pH_{b}|)$ noticeably grows due to the osmotic, $\Delta F_{vol}\left(z\right)$ contribution. Hence, below the IEP, BSA can be weakly absorbed in the central region of the brush, but is expelled from the proximity of the grafting surface.

As $|\Delta pH_{b}|$ increases ($pH_{b}$ decreases), the minimum first becomes “metastable” (with $\Delta F(z=z_{min})>0$ and eventually is converted into a quasi-plateau: the BSA is expelled from the brush. At high salt concentration, when Coulomb interactions (and re-ionization) are suppressed, the net BSA insertion free energy is dominated by a competition between osmotic repulsion and short-range attraction and the insertion free energy profiles become similar for polycationnic and polyanionic brushes. Hence, as one can see from Figure 7, at high salt concentration the BSA may be weakly absorbed by the polyanionic, as well as by the polycationic brush due to non-electrostatic interactions.

## 3. Methods and Materials

The self-consistent electrostatic potential $\psi \left(z\right)\equiv e\Psi \left(z\right)/k_{B}T$, in the negatively/positively charged (anionic/cationic) polyelectrolyte brush was derived within Poisson-Boltzmann approximation in refs. [29,30]
(17)$$\psi_{in}\left(z\right)=\pm \frac{z^{2}-H^{2}}{H_{0}^{2}}\pm 2\ln\left[\frac{\sqrt{{\left(\kappa \tilde{\Lambda }\right)}^{2}+1}-1}{\kappa \tilde{\Lambda }}\right],\;0\leq z\leq H$$
where signs “+” and “−” apply to negatively and positively charged brushes, respectively. Here, *z* is the distance from the grafting surface, *H* is the total thickness of the brush (cut-off of the polymer density profile), and $H_{0}$ is the characteristic length, defined as
(18)$$H_{0}/a=\sqrt{\frac{8}{3\pi^{2}}}N\alpha^{1/2}.$$

The electrostatic potential described by Equation (Equation 17) demonstrates continuous and smooth crossover with the electrostatic potential profile outside the brush, i.e., at z≥H, given by
$$\psi_{out}\left(z\right)=$$
(19)$$\mp 2\ln\left[\frac{\left(\kappa \tilde{\Lambda }+\sqrt{{\left(\kappa \tilde{\Lambda }\right)}^{2}+1}-1\right)+\left(\kappa \tilde{\Lambda }-\sqrt{{\left(\kappa \tilde{\Lambda }\right)}^{2}+1}+1\right)e^{-\kappa (z-H)}}{\left(\kappa \tilde{\Lambda }+\sqrt{{\left(\kappa \tilde{\Lambda }\right)}^{2}+1}-1\right)-\left(\kappa \tilde{\Lambda }-\sqrt{{\left(\kappa \tilde{\Lambda }\right)}^{2}+1}+1\right)e^{-\kappa (z-H)}}\right]$$
where signs “−” and “+” apply to negatively and positively charged brushes, respectively, and
(20)$$\tilde{\Lambda }=\frac{1}{2\pi l_{B}\left|\tilde{Q}\right|}=\frac{H_{0}^{2}}{H}$$
is the Gouy-Chapman length, which controls distribution of electrostatic potential and small mobile ions outside the brush, i.e., at z≥H, and
(21)$$\tilde{Q}=\mp \frac{1}{4\pi l_{B}}\int_{0}^{H}\frac{d^{2}\psi_{in}\left(z\right)}{dz^{2}}dz=\mp \frac{H}{2\pi l_{B}H_{0}^{2}}$$
is the residual charge per unit area of the brush, where signs “−” and “+” apply to negatively and positively charged brushes, respectively, and
(22)$$\kappa^{-1}={\left(8\pi l_{B}c_{s}\right)}^{-1/2}$$
is the Debye screening length where $l_{B}=e^{2}/\epsilon k_{B}T$ is the Bjerrum length (here *e* is the elementary charge, ϵ is the dielectric permittivity of the solvent, $k_{B}$ is the Boltzmann constant and *T* is the temperature). In the following we assume that $l_{B}=0.7$ nm and a=0.3 nm, which leads to the proportionality factor ≈60 between molar concentration and volume fraction, $C_{s}=c_{s}a^{3}$, of salt. The potential defined by Equation (Equation 19) vanishes at z→∞.

The monomer units concentration profile inside the brush at z≤H is given by
$$\alpha c_{p}\left(z\right)=$$
(23)$$\frac{1}{2\pi l_{B}H_{0}^{2}}\left[1+2{\left(\frac{\kappa H_{0}}{2}\right)}^{2}sh\left[\frac{H^{2}-z^{2}}{H_{0}^{2}}\right]+\frac{H^{2}}{2H_{0}^{2}}sh\left[\frac{H^{2}-z^{2}}{H_{0}^{2}}\right]+\frac{2H}{H_{0}}\sqrt{{\left(\frac{\kappa H_{0}}{2}\right)}^{2}+\frac{H^{2}}{4H_{0}^{2}}}ch\left[\frac{H^{2}-z^{2}}{H_{0}^{2}}\right]\right]$$
and the height of the brush *H*, i.e., the cut-off of the monomer units density profile is obtained from the conservation condition
(24)$$\int_{0}^{H}c_{p}\left(z\right)dz=N\sigma .$$

It is worth noting that the monomer units density given by Equation (Equation 23) exhibits a jump at the edge of the brush, z=H, that is,
(25)$$\alpha c_{p}\left(z=H\right)=\frac{1}{2\pi l_{B}H_{0}^{2}}\left[1+\frac{2H}{H_{0}}\sqrt{{\left(\frac{\kappa H_{0}}{2}\right)}^{2}+\frac{H^{2}}{4H_{0}^{2}}}\right]\geq 0$$

The profiles of the electrostatic potential ψ(z), both inside and outside the polyanionic brush, and the distribution of the monomer $c_{p}\left(z\right)$ density inside the brush are presented in Figure 8a,b for selected values of the salt concentration $c_{s}$. The electrostatic potential profile in the polycationic brush can be obtained as a mirror reflection of the potential curves presented in Figure 8a with respect to the z-axis.

## 4. Conclusions

We have evaluated the free energy of insertion of a BSA into a polyelectrolyte brush by taking into account three dominant contributions: (i) the re-ionization free energy, that also comprises Coulomb energy of charged amino acid residues in the electrostatic field of the brush; (ii) osmotic contribution equal to the work performed against excess osmotic pressure upon the BSA insertion into the brush and (iii) short-range attractive interactions of the non-charged area of the BSA globule surface with the brush-forming chains. The latter one is evaluated on the basis of a semi-empirical approach, developed earlier for the analysis of interaction of non-ionic nanocolloids with polymers brushes. Notably, upon calculating these three contributions to the free energy, we disregarded variations of the electrostatic field and in polymer concentration on the length scale on the order of the BAS globule size.

Our analysis is based on the previously developed theory of polyelectrolyte brushes, which enabled calculating the electrostatic potential distribution and the monomer density profile in the brush on the level of the self-consistent field Poisson-Boltzmann approximation under the conditions of dominance of electrostatic interactions in the brush (i.e., in the experimentally most relevant regime) and without an account of any structural details of the ions and the solvent. A corresponding extension of this approach could be conducted following the lines of ref. [37].

Furthermore, our theory does not account for charge-charge correlations, including electrostatically-driven adsorption of segments of the brush-forming PEs on the oppositely charged areas on the BSA globule surface. The corresponding contribution to the free energy depends on the size and shape of charged “patches” which, in turn, are controlled by local pH and may provide additional driving (attractive) force for the BSA absorption by the brush. Therefore, our calculations provide a “lower boundary” for the environmental conditions ($pH_{b},C_{s}$) under which absorption of the BSA by the brush is thermodynamically favorable. Nor do we account for the free energy of the BSA dipole moment in the inhomogeneous electrostatic field created by the brush, which may contribute at maximum on the order of $-10^{-1}k_{B}T$ close to the edge of the brush, where the gradient of the electrostatic potential ψ(z) is the steepest.

In spite of used by the theory simplifying approximations, our theoretical findings are in reasonably good agreement with the available experimental data in the literature on BSA and other globular proteins (e.g., Bovine β-lactoglobulin, Bovine Pancreatic Ribonuclease A and Lysozyme) absorption by strong and weak (pH-sensitive) polyanionic and polycationic brushes [26,27,28,38,39,40]. In particular, protein absorption by the PE brush on the respective “wrong side” of the IEP, that is, at $pH_{b}$ above pI in the case of the polyanionic brush and $pH_{b}$ below pI in the case of the polycationic brush when the protein globule carries the net charge of the same sign as the brush, was systematically studied in experiments.

The whole set of experimental data points to the key role of electrostatic interactions in the phenomena of uptake and release of the proteins by polyelectrolyte brushes. First, it was demonstrated that protein absorption by polyelectrolyte brushes is efficient at low ionic strength while an increase in the ionic strength suppresses absorption and promotes protein release from the brush. Second, an increase/decrease in $pH_{b}$ enhances protein absorption in the polycationic/polyanionic brush, respectively. Both observations are in line with the results of our theoretical analysis, presented above. Third, as evidenced by experiments [39], the strong polyanionic (poly(styrene sulfonate)) brush better absorbs BSA than the weak (poly(acrylic acid)) one under the same conditions. On the other hand, it was shown in ref. [26] that the strong polycationic (poly [2-(methacryloyloxy) ethyl] trimethylammonium chloride) brush better absorbs BSA and β-glucosidase than the weak polycationic (poly (2-aminoethyl methacrylate hydrochloride)) brush at low ionic strength, while the opposite trend is found at high ionic strength. Theoretical proof of the latter experimental finding requires an account of the effects of pH and ionic strength on the ionization equilibrium of both amino acid residues of the protein and brush forming chains, which will be carried out in our forthcoming publication.

The theory developed here does not predict a universal dependence of the protein insertion free energy on the absolute value $|pH_{b}-pI|$ for both cases of polyanionic and polycationic brushes. On the contrary, our theory predicts that the polyanionic brush more strongly absorbs BSA than the polycationic one at $pH_{b}$ close to the BSA IEP. Also the polyanionic brush can efficiently absorb BSA at larger deviation from the IEP compared to the polycationic brush, which is consistent with the experimental observations in ref. [40] where interactions of BSA and Lysozyme with quenched cationic poly ([2-(methacryloyloxy) ethyl] trimethylammonium chloride) and anionic poly (3-sulfopropyl methacrylate potassium salt) brushes were studied. This “brush charge sign asymmetry” effect predicted theoretically and observed experimentally is explained by a specific for each protein composition of weak cationic/anionic amino acid residues with respective $pK^{\prime }s$, while for other globular proteins, either the similar or the opposite trend may be found.

## Figures and Tables

**Figure 1 ijms-24-03395-f001:**
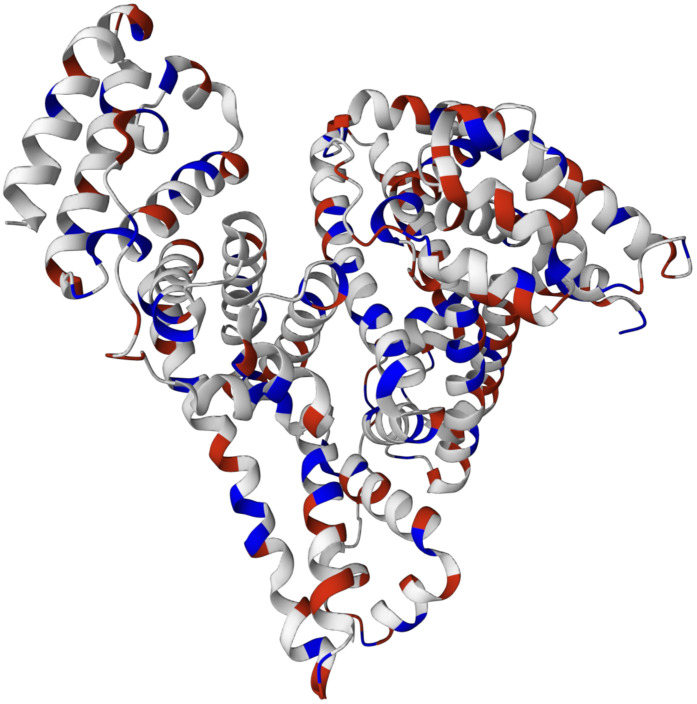
Crystal Structure of Bovine Serum Albumin [32]. Here, the blue ones are positively charged groups, the red ones are negatively charged, and the white ones are non-polar groups.

**Figure 2 ijms-24-03395-f002:**
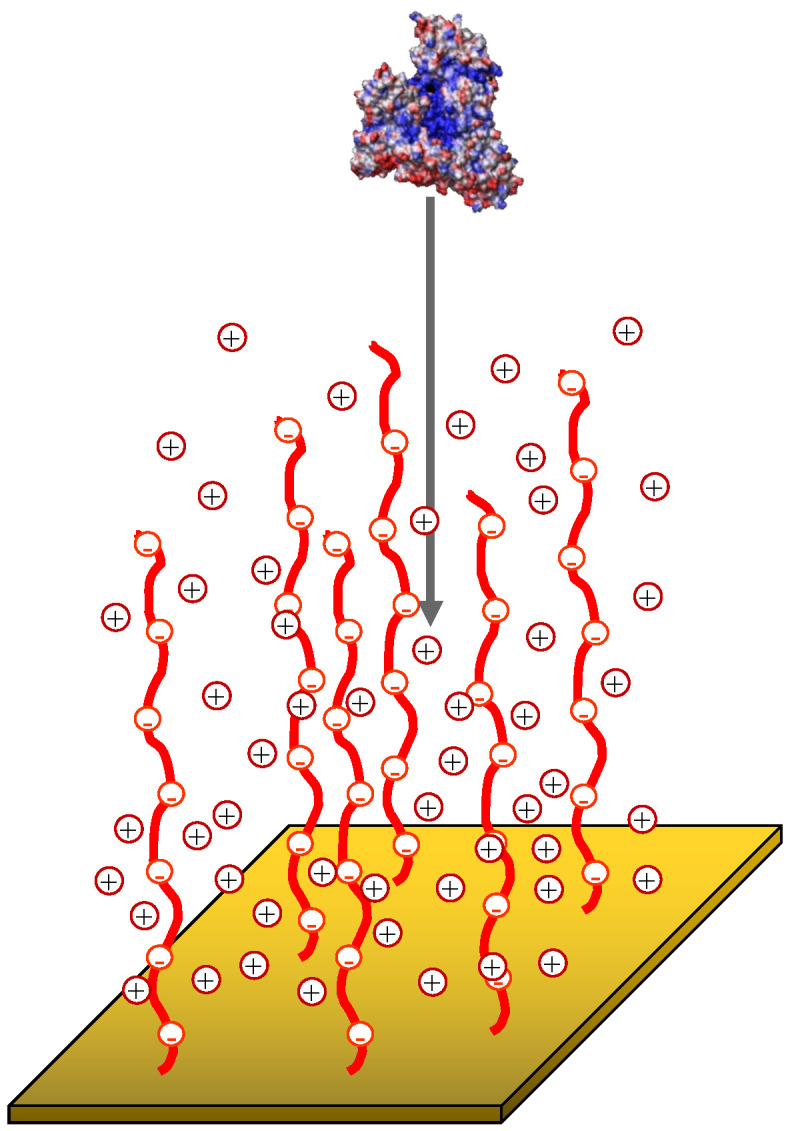
Schematics of the BSA globule insertion into the polyanionic brush. Mobile counterions (cations) are shown explicity. Blue, red and white patches on the globule surface correspond to anionic, cationic and neutral residues, respectively.

**Figure 3 ijms-24-03395-f003:**
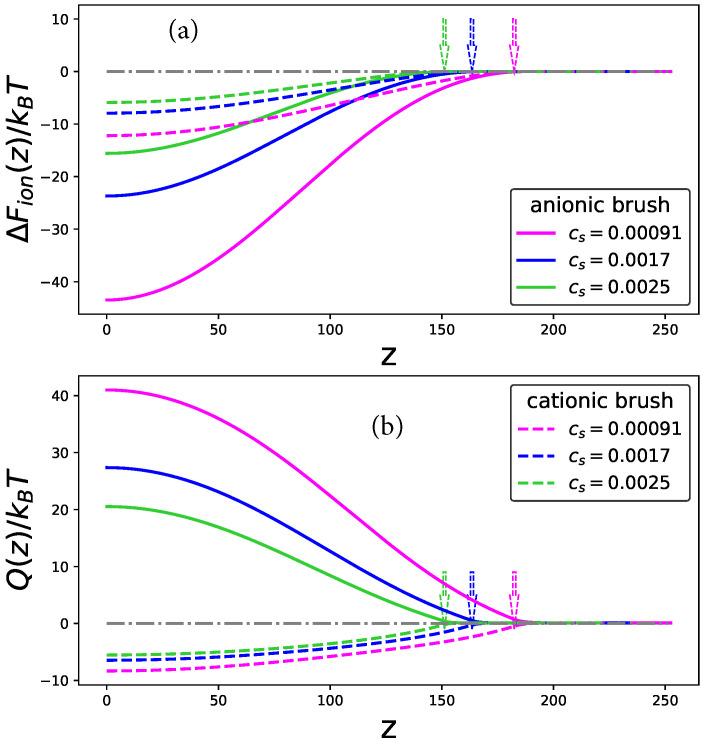
Dependences of the re-ionization free energy $\Delta F_{ion}\left(z\right)$ (**a**) and net charge of the BSA globule (**b**) on the distance from the grafting surface *z* for the cases of polyanionic (solid lines) and polycationic (dashed lines) brushes at the IEP. The color code corresponding to different salt concentrations $C_{s}$ is indicated in the legend. Other parameters are $pH_{b}=pI=5.25$, s=100, N=300. Arrows indicate the upper boundary of the brush (z=H).

**Figure 4 ijms-24-03395-f004:**
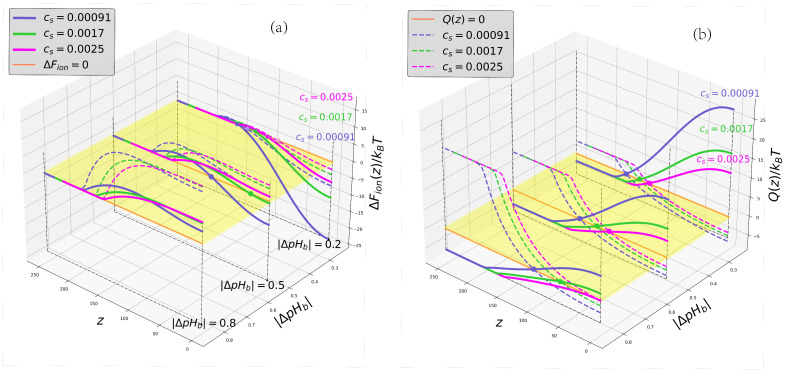
Cross-sections of the 2D profiles of the insertion free energy $\Delta F_{ion}(z,|\Delta pH_{b}\left|\right)$ (**a**) and BSA globule charge $Q(z,|\Delta pH_{b}|)$ (**b**) for the cases of polyanionic (solid curves, $pH_{b}>pI$ ) and polycationic (dashed curves, $pH_{b}<pI$) brushes. The color code corresponding to different salt concentrations $C_{s}$ is indicated at the curves. Colored circles in the panel correspond to the charge inversion points $z=z^{*}$.

**Figure 5 ijms-24-03395-f005:**
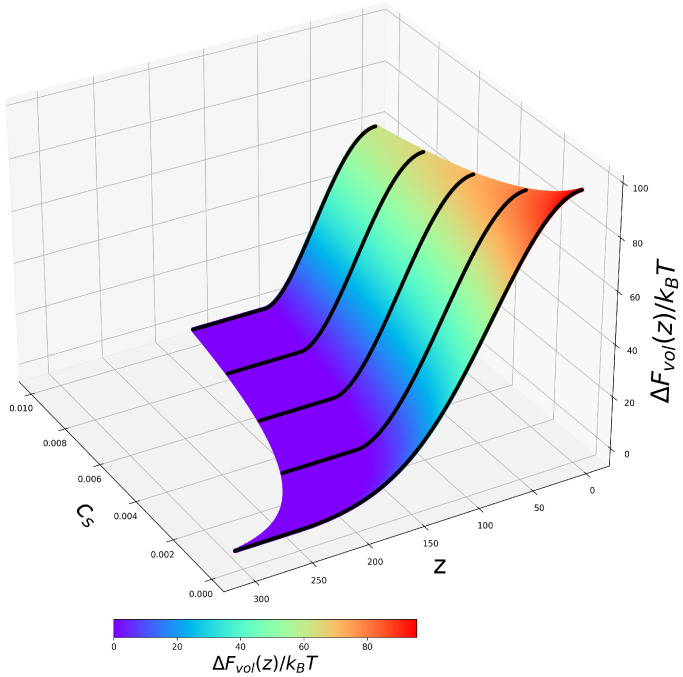
2D profiles of the osmotic contribution $\Delta F_{vol}\left(z,C_{s}\right)$ to the free energy of the BSA globule in the PE brush plotted according to Equation (Equation 14).

**Figure 6 ijms-24-03395-f006:**
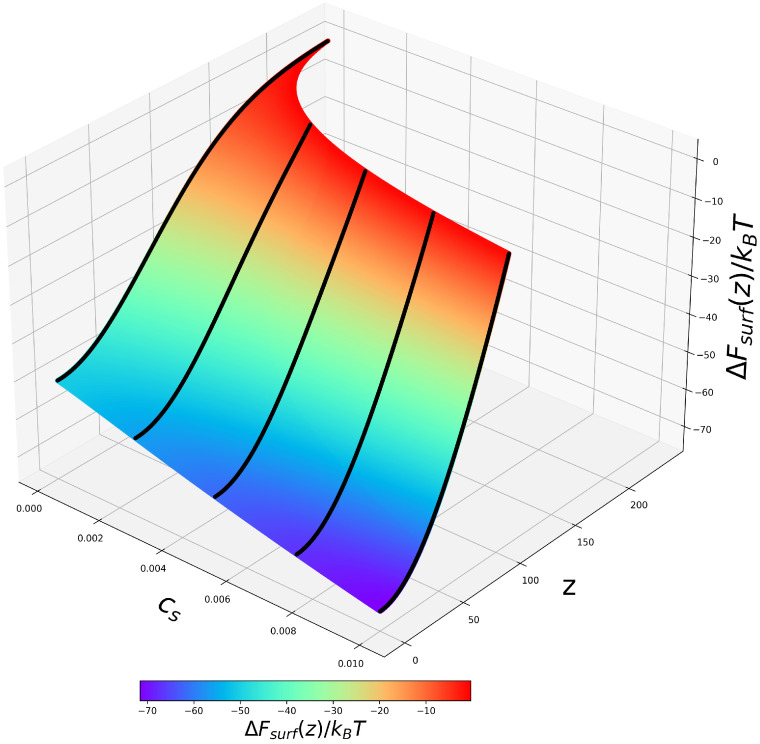
2D profiles of the free energy $\Delta F_{surf}\left(z,C_{s}\right)$ of short-range (attractive) interactions of the BSA globule with the PE brush-forming chains plotted according to Equation (Equation 15).

**Figure 7 ijms-24-03395-f007:**
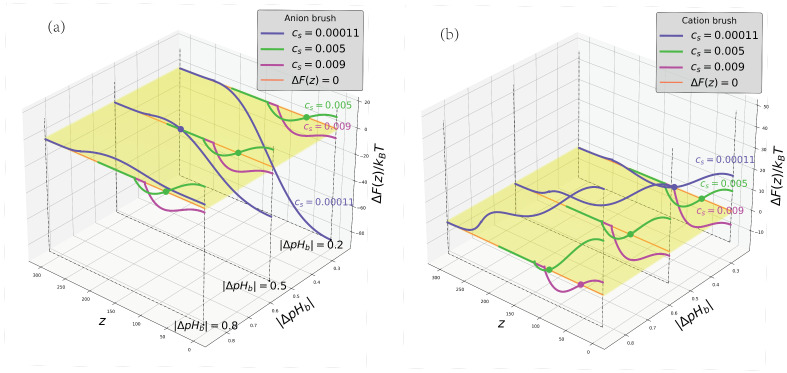
Cross-sections of the 2D profiles of the BSA insertion free energy $\Delta F_{}(z,|\Delta pH_{b}\left|\right)$, for the cases of polyanionic, $pH_{b}>pI$ (**a**) and polycationic, $pH_{b}<pI$ (**b**) brushes. The color code corresponding to different salt concentrations $C_{s}$ is indicated in the legend and at the curves. Colored circles correspond to the points of vanishing ΔF(z).

**Figure 8 ijms-24-03395-f008:**
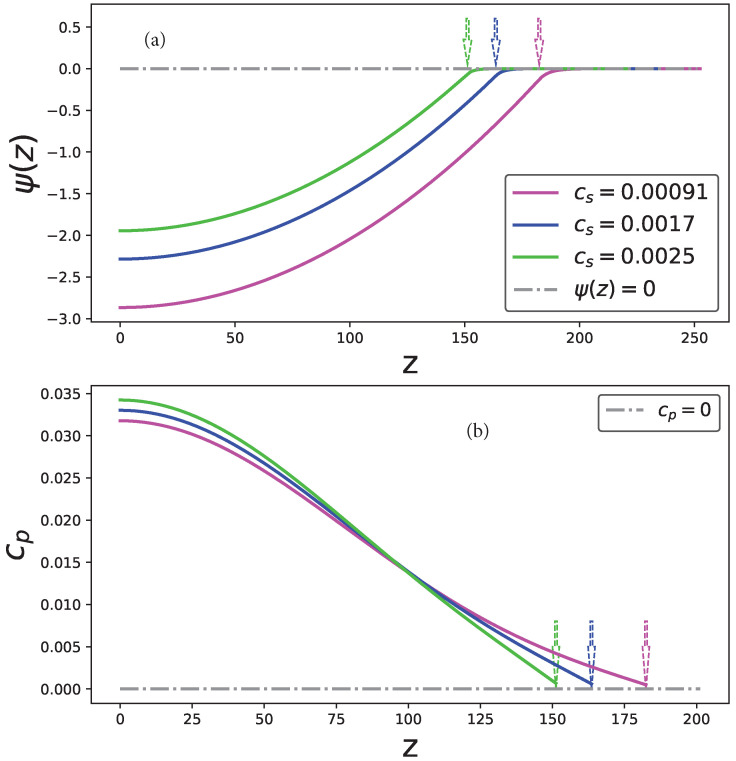
Electrostatic potential of the polyanionic brush (**a**) and the distribution of the monomer $c_{p}\left(z\right)$ density $c_{p}\left(z\right)$ inside the brush (**b**) as a function of the distance from the grafting surface *z* and varied salt concentration $C_{s}\equiv c_{s}a^{3}$ for N=300, $s/a^{2}=100$. The brush boundary, z=H, is indicated by the arrows.

**Table 1 ijms-24-03395-t001:** Values of $pK_{a}$ and numbers (N) of each charge group for BSA [35].

Amino Acid Name	N	*pK_a_*
Asp	40	3.92
Glu	59	3.92
His	16	6.9
Tyr	19	10.35
Lys	57	9.8
Arg	22	12.0
N-terminus	1	7.75
C-terminus	1	3.75

## Data Availability

Not applicable.

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
