# Peer review of "Bovine Serum Albumin Interaction with Polyanionic and Polycationic Brushes: The Case Theoretical Study"

_ijms, 2023, doi:10.3390/ijms24043395_

Round 1
Reviewer 1 Report
In the article entitled „Bovine Serum Albumin interaction with polyanionic and polycationic brushes: the case theoretical study“ by authors T. Salamatova, E. Zhulina, and O. Borisov, a fundamental study of the interaction between BSA and polyelectrolyte brushes, providing the interesting model. The issue tackled in this article is important from a fundamental perspective and the presented model will provide a significant improvement in the prediction of protein interaction. Although the theoretical approach and proposed model are highly scientifically relevant, the presentation of data, structure of the Manuscript, and most importantly detailed explanation of methodology are missing which significantly hampers the quality of the article and the possibility for the reproduction of data. My recommendation is a major revision, followed by more specific comments.
Major remarks:
- Provide detailed information about the parameters of coarse-grained model used in this article.
- It is unclear if the authors perform any kind of optimization of the initial crystal structure of BSA, or if they used crystal structure taken from the database without any further treatment. If there is no pre-optimization of BSA structure, it is mandatory to provide a sufficient explanation for this approach, since it is not common. Moreover, the structure presented in Figure 1 consists of non-BSA compounds (domain IIIB).
- Please, provide a detailed description of polymeric brushes and mobile ions. What are the structure, chemical formulas, etc? Low molecular weight salt is not sufficient.
- How authors treated the solvent molecules in their model?
- How proposed model treated the change in the protonation state of some amino acids due to the change in pH?
- Did authors consider H-bond formation in their model? If not, please, explain.
Minor remarks:
- Authors should follow the Journal guidelines, and provide the Manuscript in the desired layout.
- Please, re-write Sections 2 and 3, and include the Material and Method chapter.
- Since some references in a reference list are missing
- The formatting of Manuscript is extremely poor and needs to be significantly improved (for instance Figures should be appropriately labeled, follow the main text, etc).
Author Response
Report 1
In the article entitled „Bovine Serum Albumin interaction with polyanionic and polycationic brushes: the case theoretical study“ by authors T. Salamatova, E. Zhulina, and O. Borisov, a fundamental study of the interaction between BSA and polyelectrolyte brushes, providing the interesting model. The issue tackled in this article is important from a fundamental perspective and the presented model will provide a significant improvement in the prediction of protein interaction. Although the theoretical approach and proposed model are highly scientifically relevant, the presentation of data, structure of the Manuscript, and most importantly detailed explanation of methodology are missing which significantly hampers the quality of the article and the possibility for the reproduction of data. My recommendation is a major revision, followed by more specific comments.
Major remarks:
- Provide detailed information about the parameters of coarse-grained model used in this article.
Reply: The parameters of the coarse-grained model of the BSA are described in the first two paragraphs of the “Results and Discussion ” section. The parameters of the polyelectrolyte brush model are described in the “Materials and Methods” (after eq 22).
- It is unclear if the authors perform any kind of optimization of the initial crystal structure of BSA, or if they used crystal structure taken from the database without any further treatment. If there is no pre-optimization of BSA structure, it is mandatory to provide a sufficient explanation for this approach, since it is not common. Moreover, the structure presented in Figure 1 consists of non-BSA compounds (domain IIIB).
Reply: within our coarse-grained model the BSA is characterized by the following set of parameters: volume, surface area, numbers of specific amino acid residues of each type on the globule-solvent interface and their ionization (protonation or deprotonation) constants. All these parameters are presented in section 2 and in Table 1 and taken from crystal structure database. We anticipate that within the level of accuracy of our coarse-grained model no optimization is necessary since not all, but only the dominant interactions are accounted for. To the best of our understanding, native BSA (used in experiments on interaction with polyelectrolyte brushes) has a depicted in Figures 1 and 2 characteristic “heart” shape and comprises domains: IB, IIB, IIIB.
- Please, provide a detailed description of polymeric brushes and mobile ions. What are the structure, chemical formulas, etc? Low molecular weight salt is not sufficient.
Reply: on the level of employed here coarse-grained model the particular chemical structure of polyelectrolytes and mobile salt ions are not specified. It is essential that the mobile ions are monovalent (generalization for the case of multivalent ions is straightforward but requires specific analysis) and brush forming polyelectrolytes are strong (not pH-sensitive).
- How authors treated the solvent molecules in their model?
Reply: the solvent molecules are treated implicitly, we have added a corresponding remark in p.8 in the description of the model.
- How proposed model treated the change in the protonation state of some amino acids due to the change in pH?
Reply: The account of change of the protonation state of the amino acid residues is an essential feature of our model, which gives rise to the -re-ionization free energy, see eq 11 and the paragraph below it.
- Did authors consider H-bond formation in their model? If not, please, explain.
Reply: No, formation of hydrogen bonding (between protein and brush-forming chains) is not included in the model. This could be more relevant in the case of weak polyelectrolyte brushes which will be considered in the forthcoming publication.
Minor remarks:
- Authors should follow the Journal guidelines, and provide the Manuscript in the desired layout.
- Please, re-write Sections 2 and 3, and include the Material and Method chapter.
- Since some references in a reference list are missing
- The formatting of Manuscript is extremely poor and needs to be significantly improved (for instance Figures should be appropriately labelled, follow the main text, etc).
Reply: the manuscript is reformatted according to the recommended by the journal layout and includes Introduction, Results, Discussion, Conclusions, Materials and Methods sections

Reviewer 2 Report
Abstract is quite long and does not include crucial information about the results.
Broader insight in the problem is missing (see Incorporation of ion and solvent structure into mean-field modeling of the electric double layer. Advances in colloid and interface science, 249, 2017, 220-233.
)
I do not see any direct comparison of author’s theory with experiments
Authors should think how to take into account the charge discreteness consistently (see the reference above).
Discussion needs to be improved.
Author Response
Report 2
Comment: Abstract is quite long and does not include crucial information about the results.
Reply: Abstract is shortened and re-formulated
Comment: Broader insight in the problem is missing (see Incorporation of ion and solvent structure into mean-field modeling of the electric double layer. Advances in colloid and interface science, 249, 2017, 220-233.)
Reply: we are thankful to the reviewer for attracting our attention to this interesting paper which we cite now as ref 42. We anticipate, however, that our model accounts for the dominant interactions that govern absorption of a weak polyampholytic nanoparticle (e.g., BSA) by a polyelectrolyte brush. As an essential pre-requisite of the developed theory we use the analytical expression for the electrostatic potential created by the polyelectrolyte brush which is available only in the limiting case of dominance of Coulomb intermolecular interactions acting between brush-forming chains. Incorporation of, e.g., excluded volume short-range interactions in the self-consistent field Poisson-Boltzmann framework is straightforward, but makes impossible obtaining the profile of the electrostatic potential in the explicit analytical form. Moreover, account for structural ion and solvent properties require implementing of the additional model assumption and non-universal parameters thus diminishing the generality of our results.
Comment: I do not see any direct comparison of author’s theory with experiments
Reply: We have added a paragraph in the “Conclusions” comparing our theoretical findings with available experimental data. Unfortunately, at present in the literature one finds only data concerning the BSA interaction with strong and weak either polyanionic or polycationic brushes. No systematic comparison of the results obtained for architecturally similar strong polyanionic and polycationic brush is available at present and this is the central point of our theoretical analysis. We stress that we do not aim to reproduce quantitatively any experimental result for a brush with particular chemical structure but rather predict general qualitative effect that should be observed in future experiments irrespectively of the specific chemistry.
Comment: Authors should think how to take into account the charge discreteness consistently (see the reference above).
Reply: We have included suggested by the reviewer paper in the reference list (ref 42) and explained in the text that we use a minimal model that enables us to study the interplay and competition of electrostatic and short-range hydrophobic interactions in the interaction of BSA with a polyelectrolyte brush.
Comment: Discussion needs to be improved.
Reply: Discussion is improved.

Round 2
Reviewer 1 Report
Authors improved Manuscript significantly, therefore my recommendation is to accept the paper in present form.
Reviewer 2 Report
can be accepted